# PV System Failures Diagnosis Based on Multiscale Dispersion Entropy

**DOI:** 10.3390/e24091311

**Published:** 2022-09-16

**Authors:** Carole Lebreton, Fabrice Kbidi, Alexandre Graillet, Tifenn Jegado, Frédéric Alicalapa, Michel Benne, Cédric Damour

**Affiliations:** Energy Lab, Université de La Réunion, 15, Avenue René Cassin CS 92003, CEDEX 9, 97744 Saint-Denis, France

**Keywords:** diagnosis, PV system, variational mode decomposition, multiscale entropy

## Abstract

Photovoltaic (PV) system diagnosis is a growing research domain likewise solar energy’s ongoing significant expansion. Indeed, efficient Fault Detection and Diagnosis (FDD) tools are crucial to guarantee reliability, avoid premature aging and improve the profitability of PV plants. In this paper, an on-line diagnosis method using the PV plant electrical output is presented. This entirely signal-based method combines variational mode decomposition (VMD) and multiscale dispersion entropy (MDE) for the purpose of detecting and isolating faults in a real grid-connected PV plant. The present method seeks a low-cost design, an ease of implementation and a low computation cost. Taking into account the innovation of applying these techniques to PV FDD, the VMD and MDE procedures as well as parameters identification are carefully detailed. The proposed FFD approach performance is assessed on a real rooftop PV plant with experimentally induced faults, and the first results reveal the MDE approach has good suitability for PV plants diagnosis.

## 1. Introduction

In this period of ecological and environmental awareness, renewable and natural energy sources have experienced an important generalization and development in the last 20 years. Territories with a vast potential for solar energy boost their photovoltaic production and expand the number of solar plants. The stakes are even higher for non-interconnected island territories, such as the overseas territories of France (Corsica, etc.).

Favorably located in an intertropical zone, La Réunion increased its PV energy production from 0 to 206.3 MW in 15 years [1]. The La Réunion grid has the particularity to be non-interconnected with electrical production that is mainly imported fossil fuel-based. The growing proportion of intermittent energy such as PV sources increases the risk of a strong variation in energy production. In order to ensure the stability of this insular power grid, the need for maintenance and energy production forecast increases with the size of the PV park. In addition to the normal aging of the oldest plants, humid tropical climate conditions induce an increase in faults and premature aging of the PV systems [2]. Moreover, La Reunion’s PV park includes plants in a wide range of sizes, with a large proportion of small installations, for example, 62.66% of the number of the PV plants are below 9 kVA, and are unevenly equipped for FDD.

PV systems can be prone to degradation faults and electrical faults with different occurrences and severity [3,4,5]. The meteorological conditions highly impact the PV aging velocity, as a hot-humid climate and thermal cycling have a negative effect on PV modules [6,7]. The conventional detection and protection methods available in the industry are already used in PV plants and allow to avoid electrical faults such as line–ground (L-G) faults, line–line (L-L) faults, arc series faults and open-circuit (OC) faults [8,9]. They are electrical fuses, circuit breakers, surge protectors [10]. Arc fault detectors are not mandatory on EU installations, and their installation is expensive.

Degradation faults are hotspots, cell cracks, discoloration, delamination, connections sectioning, bypass diode dysfunction, mismatch, corrosion, shading or potential induced degradation (PID) [2,11,12]. PV degradation faults detection requires the setting-up of methods such as electroluminescence, infrared thermography, UV fluorescence and I–V tracers. These methods have an important cost [10], particularly in the case of small installations, and have the disadvantage of being on demand at the moment that a drop in production has already been noticeable. Furthermore, they involve the deployment of human and technical resources and are complex to implement in the case of rooftop plants. Some of them require module disconnection and therefore a production loss. Nevertheless, technics are newly developed to allow a rapid module analysis as an ultra-rapid and low-cost I–V curve [13], containing an additional acquisition system with a high sampling frequency.

To solve these issues, it is important to enable the deployment of low-cost, on-line, easy-to-implement FDD solutions that target all production plant dimensions. The DETECT (Diagnosis onlinE of sTate of health of EleCTric systems) research project takes root in this context, with the objective to develop an on-line and low-cost FDD with no additional sensors or costly equipment. PV FDD is crucial and has taken a lot of effort in the last 10 years, and a considerable amount of research articles have been published. Reviews list and demonstrate the extent of the great diversity of the developed methods [11,14,15,16,17,18,19].

In addition to the usual components of protection [9] and visual solutions [20], advanced fault detection methods are developed to enhance FDD and increase the PV system reliability. Four types of advanced FDD can be distinguished:A model-based diagnostics method;Real-time measurements;An Output Signal Analysis (OSA);A machine learning-based diagnosis.

According to [18], the Output Signal Analysis (OSA) assumes that any fault has an effect on the PV system output. The output current or voltage can be affected by a drop in values or a change in signal dynamics. A fitted analysis of these perturbations allows the extracting diagnostics behavior of each fault. An OSA takes the advantages to require a limited number of sensors which make it low-cost. With the accuracy of the detection and the low complexity of the installation, an OSA is a prime candidate for the on-line and real-time diagnosis. An OSA-based diagnosis remains noise-sensitive, and the recent and ongoing research aims to reduce the impact of noise on the detection capability using a novel approach [21].

Then, different methods coming from cross-domain applications have been implemented for PV system fault diagnosis. Classification tools have already widely been tested to detect PV faults, such as Decision Tree [22], the Probabilistic Neural Network classifier [23], Random Forest [24], the Artificial Neural Network classifier [25], a One-class Support Vector Machine [26] and Machine Learning Based on Gaussian Process Regression [27]. Machine learning has been widely experienced these last years. Review papers list and compare the different machine learning technic in the case of PV fault diagnosis [28,29].

The method limitations are the learning data and preliminary analysis needs, which limits the FDD robustness and scalability.

Decomposition techniques have already been tested in PV FDD, particularly Wavelet Transform. Undecimated Wavelet Transform is used to detect PV power plant swag and swell [21]. Wavelet Transform associated to Radial Basis Function Networks (RBFNs) has been applied to early failure detection and fault classification on a real PV plant [30]. Multiscale wavelet decomposition is applied to achieve a non-sensitive-to-noise short-circuit, open-circuit and shading FDD [31]. A statistical analysis such as Principal Component Analysis was performed for a shading diagnosis [32]. These decomposition algorithms require complex functions, which limits the implementation convenience and increases the computational cost. Novel decomposition algorithms have been developed, as mentioned below. Empirical Mode Decomposition (EMD) is a recursive algorithm that decomposes a signal into several modes of separate spectral bands, named Intrinsic Mode Functions (IMFs). This signal processing tool was introduced in 1998 [33] and is widely used in the diagnosis domain. In the case of PV faults, EMD is applied to transform a non-stationary time series into a stable time series that allows a comparison between a model and a real performance in order to establish a state of health index [34]. The output current is decomposed by EMD, and a classification tool based on machine learning is added to localize the islanding and tripping of the distributed generation [35]. EMD limitations are also well-known, which are the ending effect, mixing mode in non-stationary signals and sensitivity to noise and sampling. To overcome these limitations, variational mode decomposition (VMD) has been developed [36]. Power quality disturbances in a Grid-Connected Distributed Generation System are detected using VMD [37]. In two cases of EMD and VMD [35,37], fault identification is carried out by Decision Tree classification.

Among the important diversity of the methods, information theory offers innovative and accurate tools. Already present in other fields of research, entropy is widely used in medical applications and commonly used in other diagnosis applications. [38] initiates the use of entropy in PV fault detection, to the specific case of arc fault detection. Sample entropy (SE) is performed to detect L-L, L-G, OC, weather disturbances, high impedance, blocking diode and partial shading faults [39]. However, the faults are not isolated and localized. These works encourage drawing inspiration from other FDD application domains, taking advantage of entropy efficiency. Multiscale entropy is increasingly used in diagnosis research fields such as in biomedical time series, electroseismic time series, rotary machine vibrations and financial time series [40,41]. A multiscale approach allows evaluating entropy of different phenomena with different time scales, and the computation of global entropy is the sum of each time scales’ entropy. In order to analyze a biomedical time series involving different time scales, multivariate and multiscale dispersion entropy is developed [42].

The superposition of VMD and different types of entropy computation are studied, such as multiscale dispersion entropy (MDE) to diagnose partial discharge [43], Modified Multiscale Symbolic Dynamic Entropy or Generalized Composite Multiscale Symbolic Dynamic Entropy for planetary gearboxes fault diagnosis [44] and, more recently, VMD is associated to Improved Multiscale Fuzzy Entropy to PV arc fault detection [45]. The developed methods, even with real-time applicability, require a high sampling frequency and thus an addition of a sample device on a standard PV plant. Furthermore, the methods cannot be extended to all time scale faults.

On this basis, the presented work takes advantage of the techniques named above in order to detect the faults presented on a PV installation. VMD is applied to extract information-carrying signals, removing the non-informative noise with an IMF’s selection criterion. In addition, an MDE analysis is performed in order to identify PV faults. The proposed method tackles the machine learning limitations with no learning dataset, for the purpose of good flexibility and robustness. The proposed decomposition algorithm has the advantage of an ease of implementation, conversely to other decomposition tools such as WT. The multiscale approach will be used, allowing detecting faults with different time scales.

The proposed FDD requires only an additional software component. Using the electrical data commonly collected by the inverters, the FDD is easy to implement in new or existing PV plants. The algorithm has been tested on a real rooftop PV plant of 4 kW. The advantages of the proposed method are as follows:Model identification is not necessary;There is no dependence on the PV plant characteristics;It is insensitive to weather variations;It has a low computation cost.

The present work aims to synthetically present the used method and to clearly and methodically group all of the elements allowing rapid application and repeatability. Furthermore, the first results of the proposed method applied to the PV faults diagnosis are shown. This is, as far as we know, the first application of this method to shading faults.

The paper is organized as follows: Section 2 introduces VMD, the MDE concept description and the proposed fault diagnosis strategy. Section 3 presents the experimental setup and the data used to develop and validate the proposed method. Section 4 is dedicated to the experimental results obtained using the proposed diagnosis method. The conclusion is provided in Section 5.

## 2. Methods

### 2.1. VMD

VMD, as with EMD, is an algorithmic approach that demodulates an original signal x(t) into a finite number K of amplitude-modulated–frequency-modulated (AM-FM) signals $x_{k}\left(t\right)$, called Intrinsic Mode Functions (IMFs). Adaptive, very efficient and based on the signal local characteristics, they are applicable to non-linear and non-stationary signals [36].
(1)$$x\left(t\right)=\sum_{k=1}^{K}x_{k}\left(t\right)=\sum_{k=1}^{K}A_{k}\left(t\right)\;.\;cos\phi_{k}\left(t\right)$$

$x_{k}\left(t\right)$ fulfills the following conditions:The number of local extrema and zero-crossing differs at most by one.The average of the upper and lower envelope defines local maxima and local minima, respectively, zero at any point.

The primary difference between EMD and VMD is the identification method of each IMF. EMD is a recursive algorithm, IMFs are identified successively until a stopping criterion has been reached, while VMD is entirely non-recursive, and IMFs are simultaneously defined. In this work, VMD is retained to perform the signal decomposition.

VMD aims to solve the following constrained variational problem, obtaining the minimal bandwidth of each mode:(2)$$min{\;}_{x_{k},\omega_{k}}\left\{\sum_{k}{∥\delta_{t}\left[\left(\sigma \left(t\right)+\frac{j}{\pi t}\right)*x_{k}\left(t\right)\right]e^{-j\omega_{k}t}∥}_{2}^{2}\right\}$$
where $\left\{\omega_{k}=\omega_{1},\;\omega_{2},\;\ldots ,\;\omega_{K}\right\}$ the central frequencies of each mode, σ is the Dirac distribution and $\delta_{t}$ the derivation of *t*. Central frequencies and bandwidth of each mode are determined by searching the optimal solution of the variational model. This equation is solved by Alternate Direction Method of Multipliers (ADMM), using Lagrangian multipliers and quadratic penalty terms:(3)$$\begin{array}{cc} \Gamma \left(x_{k},\omega_{k},\lambda \right)= & \;\alpha \sum_{k}{∥\delta_{t}\left[\left(\sigma \left(t\right)+\frac{j}{\pi t}\right)*x_{k}\left(t\right)\right]e^{-j\omega_{k}t}∥}_{2}^{2}+{∥x\left(t\right)-\sum_{k}x_{k}\left(t\right)∥}_{2}^{2}\\ \; & +\langle \lambda \left(t\right),x\left(t\right)-\sum_{k}x_{k}\left(t\right)\rangle \end{array}$$
where α is the quadratic penalty factor, and λ represents the Lagrangian multiplier. The last equation is solved in frequency domain Parseval/Plancherel Fourier isometry and Hermitian symmetry [46] and results in following equations:(4)$$x_{k}^{j+1}=\left(\hat{x}-\sum_{i\neq k}\hat{x_{i}}+\frac{\hat{\lambda }}{2}\right)\frac{1}{1+2\alpha {\left(\omega -\omega_{k}\right)}^{2}}$$
(5)$$\omega_{n}^{j+1}=\frac{\int_{0}^{\infty }\omega {\left|x_{k_{n}}\left(\omega \right)\right|}^{2}\;d\omega }{\int_{0}^{\infty }{\left|x_{k_{n}}\left(\omega \right)\right|}^{2}\;d\omega }$$
The symbol $\hat{\;}$ denotes the frequency domain variables. The different steps of VMD are described as follows:Step 1

*n* = 1. Initialization of $\left[{\hat{x_{k}}}^{1}\right]$, $\left[\omega_{k}^{1}\right]$ and of ${\hat{\lambda }}^{1}$, spectrum and center frequencies of each k mode, at iteration number 1.

Step 2

*n* + 1. For all k mode, $\left[{\hat{x_{k}}}^{n+1}\right]$ is updated
(6)$${\hat{x}}_{k}^{n+1}=\frac{\hat{x}-\sum_{i<k}{\hat{x}}_{i}^{n+1}-\sum_{i>k}{\hat{x}}_{i}^{n}+\frac{{\hat{\lambda }}^{n}}{2}}{1+2\alpha {\left(\omega -\omega_{k}^{n}\right)}^{2}}$$
where α is the bandwidth parameter. $x_{k}\left(t\right)$ is obtained taking the real part of Inverse Fourier Transform of $\hat{x_{k}}$.

Step 3

$\left[\omega_{k}^{n+1}\right]$ is updated.
(7)$$\omega_{k}^{n+1}=\frac{\int_{0}^{\infty }\omega {\left|{\hat{x}}_{k}\left(\omega \right)\right|}^{2}\;d\omega }{\int_{0}^{\infty }{\left|{\hat{x}}_{k}\left(\omega \right)\right|}^{2}\;d\omega }$$

The center frequency is updated at the center of gravity of the power spectrum of corresponding mode [37].

Step 4

${\hat{\lambda }}^{n+1}$ is updated
(8)$${\hat{\lambda }}^{n+1}={\hat{\lambda }}^{n}+\tau \left(\hat{x}-\sum_{k}{\hat{x_{k}}}^{n+1}\right)$$
where τ update parameter.

Step 5

Steps 2 to 4 are started until the tolerance value ϵ is reached:(9)$$\sum_{k}\frac{{∥{\hat{x}}_{k}^{n+1}-{\hat{x}}_{k}^{n}∥}_{2}^{2}}{{∥{\hat{x}}_{k}^{n}∥}_{2}^{2}}<\epsilon$$

### 2.2. VMD Parameters Selection

VMD algorithm parameters have to be defined, such as τ, α and K [37].

K: In case of known number of components in the signal, it is recommended assigning this number to K. Other modes can be added and affected by noises. In most cases, the numbers of different occurring components are unknown and another analysis has to be conducted. The central frequency of each IMF is a good indicator to determine the most effective value of K. In case of excessive decomposition, the central frequencies of the last IMFs become similar [43]. A preliminary study leads to determine the maximum number of decomposition K. In case K does not correspond to the real number of signal components, the quality of the decomposition depends on the value of α.

α: is the bandwidth parameter. The value of α determines the modes’ central frequencies range. The different frequencies of the signal are assumed to be known, and in case of a wide range of frequencies, the bandwidth parameter has to be kept at a small value in order of hundreds. Conversely, to constrain the IMFs central frequencies in a small range, the value of α will be set at a higher value in the order of tens of thousands.

τ: is the update rate of the Lagrange multiplier. It influences the optimization problem convergence. In fact, the convergence velocity will be increased with a higher value of τ. However, the optimization problem can be blocked in a local extrema. τ value can be set by default at 0.01.

The different values of K, α and τ have to be determined in accordance with the system and the faults to be detected.

### 2.3. IMFs Selection

The selection of the most informative IMFs can be achieved in different ways. First of them is Relative Mode Energy Ratio (RMER) computation, which evaluates the energetic contribution of each IMF regarding original signal energy.
(10)$$RMER_{k}=\frac{\sum_{n}x_{k}^{2}}{\sum_{k}\sum_{n}x_{k}^{2}}$$

Other variables can be observed to determine the most important IMFs. For power quality disturbances detection [37], the criteria applied are the Mode Frequency bandwidth, the number of zero crossing and the instantaneous amplitude. In case of planetary gearboxes health conditions identification [44], the IMFs with the central frequency close to the gear mesh harmonics are retained, and among those, the IMF with higher central frequency is selected. In another example of partial discharge detection [43], the authors evaluate the Correlation Coefficients (CC) between each IMF and the original signal. A threshold is applied to keep the IMFs with the higher CC.

### 2.4. Multiscale Dispersion Entropy

#### 2.4.1. Introduction about Entropy

Entropy analysis is an informative theory technique introduced by Shannon [47]. Entropy is a statistical approach, quantifying the probability that neighboring points in a time series will be within a predetermined range [48]. Thereby, it measures the complexity and the richness of information of a time series. Different types of entropy computations have been developed, approximate entropy (ApE) and sample entropy (SE) [49], permutation entropy (PE) [50] and, among the latest developed, dispersion entropy (DE) [51]. SE is powerful and efficient, but its computation cost is important and depends on the time-series length. PE is an efficient irregularity indicator and is based on the order relations among values of the signal. Even if PE is simpler and computationally fast, it does not consider the mean value of amplitudes and differences between amplitudes. DE is developed to overcome SE and PE limitations. According to Shannon, the entropy *E* is defined as follows:(11)$$E=-\sum_{i=1}^{n}p_{i}logp_{i}$$

With $p_{i}$, the probability of a system being in the cell *i*. The procedure to compute $p_{i}$ in case of dispersion entropy is described in the following subsection.

#### 2.4.2. Dispersion Entropy

According to Rostaghi et al. [51], the four steps of the dispersion entropy algorithm are described below. An example is provided at each step.

Step 1.Classes mapping

In this work, the original signal is mapped with a Normal Cumulative Distribution Function (NCDF): $x=\left\{x_{1},x_{2},\ldots ,x_{N}\right\}\to y=\left\{y_{1},y_{2},\ldots ,y_{N}\right\}$ with y values from 0 to 1. An algorithm is applied in order to assign to each value of $y_{i}$ an integer value from 1 to *c*, with *c* the number of classes.
(12)$$z_{j}^{c}=round\left(c.y_{i}+0.5\right)$$

Example with *N* = 10;
x={6.787;7.577;7.431;12.200;6.555;1.712;7.061;0.318;2.769;0.462}
y={0.653;0.725;0.712;0.965;0.630;0.175;0.679;0.097;0.255;0.104}
with *c* = 3;
$$z_{j}^{c}=\left\{2\;;\;3\;;\;3\;;\;3\;;\;2\;;\;1\;;\;3\;;\;1\;;\;1\;;\;1\right\}$$

Step 2.Embedding vector and dispersion patterns creation

Embedding vectors are created, with length of m, and taking into account the time delay *d*.
$$z_{i}^{m,c}=z_{i}^{c},z_{i+d}^{c},\ldots ,z_{i+(m-1)d}^{c}\;\;,\;\;i=1,2,\ldots ,N-\left(m-1\right)d.$$
$z_{i}^{m,c}$ is mapped to a dispersion pattern $\pi_{\nu_{0}\nu_{1}\ldots \nu_{m-1}}$ where $z_{i}^{m,c}=\nu_{0},z_{i+d}^{m,c}=\nu_{1},\ldots ,z_{i+(m-1)\;\;d}^{m,c}=\nu_{m-1}$. The number of possible patterns is equal to $c^{m}$.

With *d* = 1 and *m* = 2, there will be $c^{m}=3^{2}=9$ possible dispersion patterns π: $\pi_{11}$, $\pi_{12}$, $\pi_{13}$, $\pi_{21}$, $\pi_{22}$, $\pi_{23}$, $\pi_{31}$, $\pi_{32}$, $\pi_{33}$, and N−(m−1)d=10−(2−1)×1=9 embedding vectors with a length of m=2.
$$z_{1}^{2,3}=2,3\to \pi_{23},\;z_{1}^{2,3}=2,3\to \pi_{23},\;z_{2}^{2,3}=3,3\to \pi_{33}$$
$$z_{4}^{2,3}=3,2\to \pi_{32},\;z_{5}^{2,3}=2,1\to \pi_{21},\;z_{2}^{1,3}=3,3\to \pi_{33}$$

Step 3.Relative frequency computation

The relative frequency of each pattern is defined as the number of occurrences of each pattern divided by the total number of embedding vectors:(13)$$p\left(\pi_{\nu_{0}\nu_{1}\ldots \nu_{m-1}}\right)=\frac{Number\left\{i∥i\leq N-\left(m-1\right)d,z_{i}^{m,c}\;has\;type\;\pi_{\nu_{0}\nu_{1}\ldots \nu_{m-1}}\right\}}{N-(m-1)\;d}$$

For example,
$$\begin{array}{cccccc} \; & p(\pi_{23})=1/9; & \; & p(\pi_{33})=2/9; & \; & p(\pi_{32})=1/9;\\ \; & p(\pi_{13})=1/9; & \; & p(\pi_{11})=2/9; & \; & p(\pi_{31})=1/9;\\ \; & p(\pi_{12})=0/9; & \; & p(\pi_{22})=0/9; & \; & p(\pi_{21})=1/9; \end{array}$$

Step 4.Entropy computation based on Shannon’s definition of entropy:


(14)
$$DE\left(x,m,c,d\right)=-\sum_{\pi =1}^{c^{m}}\pi_{\nu_{0}\nu_{1}\ldots \nu_{m-1}}ln\left(\pi_{\nu_{0}\nu_{1}\ldots \nu_{m-1}}\right)$$


For the example, DE(x,2,3,1)=1.8892.

In the present work, in order to enable value comparison, dispersion entropy value is normalized thanks to following equation:(15)$$DE_{norm}=\frac{DE}{ln\left(c^{m}\right)}$$

#### 2.4.3. DE Parameters Selection

The DE parameters have to be determined: *m*, *c* and *d*. A list of recommendations [51] must be taken in consideration:*d* should be set at 1 in order to avoid aliasing effects.$c^{m}$ must be smaller than the length of the time-series length.*c* must be larger than 1 to have different patterns. If *c* is too small, very different values can be assigned to the same pattern, and if *c* is too large, the DE will be sensitive to noise.*c* can be chosen from 4 to 8.*m* has to be carefully selected; with a too small value, the dynamics change will not be detected, and small variations will not be detected with a large value of *m*.The higher *m* and *c* value are, the longer the computation time will be.

### 2.5. Multiscale Approach

Multiscale approaches take into account the different time scales of the different phenomena involved in an output signal. To accomplish this, a coarse-graining procedure is applied to the original time series to create a novel time series for each scale of a length of $N/\tau_{cs}$.
(16)$$y_{j}^{\left(\tau_{sc}\right)}=\frac{1}{\tau_{sc}}\sum_{i=\left(j-1\right)\tau_{sc}+1}x_{i},1\leq j\leq N/\tau_{sc}.$$

Then, dispersion entropy is computed for each coarse-grained time series. The coarse-grained time series length decreases as time-scale ratio increases [40], as illustrated in Figure 1 for scale factor of 2 and 3. As recommended, the time-series length must be in the range $10^{m}$ to $20^{m}$ [49] to obtain a reliable entropy value.

### 2.6. Proposed Approach

The proposed method is detailed in the following flowchart in Figure 2. Inverter output current is collected and decomposed by VMD. The main important IMF is retained to compute multiscale dispersion entropy. The VMD-MDE curve trends are observed.

## 3. Experimental Setup

The data experiments are performed on a 20-year-old photovoltaic plant, installed on the University of La Réunion rooftop. The installation is composed of 2 strings of 12 modules. Each line consists of 2 parallel groups of 6 panels in series. The TE1500 and TE1700 modules installed have similar characteristics. They are composed of 72 solar cells connected in series with groups of 18 cells equipped with a bypass diode. The panels are tilted at 26° to the ground and oriented toward the north. The calculated power of line 1 is 2.04 and 2.09 kWp for line 2. Shading tests are performed on line 1. The technical characteristics of the modules are specified in Table 1.

In order to assess experimental conditions, a radiometric station has been installed on the PV plant using (Table 2):A Delta-T Devices SPN1 pyranometer installed with the same inclination and orientation and inclination angle as the PVs;A type-K thermocouple for ambient temperature;A type-T thermocouple stuck to the center of a PV to measure its temperature.

A thermal camera Flir C5 is used to regularly check the temperature of the panels during the experiments. Figure 3 shows the experimental setup for panel partial shading.

### Data Communication

Figure 4 shows a simplified diagram of the data acquisition process. To enable transformed and secure data transmission, this data pipeline consisting of gateways, network services, a database to store the data and application servers has been implemented.

Electrical data measurements are acquired by an operating and management system for photovoltaic plants connected to the electrical network, coupled with inverters. The frequency of data repatriation has been selected from the telemetric control of the software.

All radiometric and meteorological sensors are connected to a Campbell Scientific CR1000X data logger for data collection. Data transmission to the server is ensured via TCP/IP protocol. In order to exploit data in real time, data are stored with sampling frequency of 1 Hz.

To ensure the availability of data in real time, software is written in Go programming language. It monitors the data files from the various sources and injects data in the form of a time series into the InfluxDB database.

Accordingly, an end-to-end data pipeline that relies on open source technologies has been developed. Data from sensors and inverters are periodically sent, collected and stored in a centralized database system based on Go program for real-time data streaming and InfluxDB for data storage. A set of customized dashboards has been created using new features available since the last major InfluxDB, in order to update for real-time monitoring, data visualization and status control of the devices.

## 4. Results

### 4.1. Variational Mode Decomposition

The parameters for the VMD computations have been selected using the previously exposed conditions. α is set at 10,000 and τ at the value of 0.01. The number of modes K is defined in the following subsection.

#### Number of Mode Selection

The number of modes is selected thanks to the central frequencies $\omega_{c}$ observations. The VMD is performed for K = 1 to K = 8, and $\omega_{c}$ is listed in the following tables. Table 3 refers to the $\omega_{c}$ with healthy conditions data shown in Figure 5.

Table 3 exhibits that central frequencies become very close from a value of 5 modes. Therefore, for the present approach, K is defined at 5. All the VMD parameters are summarized in Table 4. The VMD result for the previously shown dataset (Figure 5) is shown in Figure 6.

### 4.2. IMFs Analysis and Selection

The analysis of the energy contribution, central frequency and correlation coefficients have been conducted on each mode (Figure 7b) and for several experimental data in order to exhibit the reactivity of a mode to different experimental conditions (Figure 7a). This first approach shows that the mode number 5 energetic contribution is higher than the other modes. However, its central frequency does not change among the experimental conditions. In this study to complete the IMFs selection, it is proposed to compute multiscale dispersion entropy on each of the five modes in the next section.

It is worth noting that a comparison between EMD and VMD has been conducted and partially published [52]. A fuller comparison between the EMD and VMD performance for a PV diagnosis is in progress.

### 4.3. Multiscale Dispersion Entropy

The time delay *d*, the number of classes *c* and the length of the embedding vector are defined in accordance with recommendations revealed in Section 2.4.3. The definition of the maximum scale factor $\tau_{s}c$ and the length of the time series *N* is an outcome of the following considerations:The size of the time series should be sufficiently short to minimize the amount of necessary data as well as the computation time and to also maximize the monitoring frequency. A window width of a maximum of 30 min is retained.The studied PV plant acquisition system frequency rate is substantially low, and in line with the DETECT project requirements, no external measurement device has been added. During 30 min, approximately 300 data points are collected.Dispersion entropy requires a minimum of points to ensure a valid value of entropy. Compliant with Section 2.4.3 and taking into account the length of the coarse-grained time series, $N/\tau_{s}c$ must be significantly higher than $c^{m}=6^{2}=36$ points.

An appropriate balance has been found and the monitoring window duration has been defined at 36 min with 360 points, and $\tau_{s}c$ has been set at 7. All of the multiscale dispersion entropy parameters are listed in Table 5.

The experimental datasets shown in Figure 8f are considered. Each segmented time series is decomposed by VMD following Table 4. The multiscale dispersion entropy is then computed in accordance with Table 5 for each IMF, resulting in Figure 8a–e. For a better understanding of the figures, the results from the no-fault clear sky data are represented in blue dashed curves, no-fault cloudy sky data in black solid curves and faulty data are displayed in red solid curves. Orange curves with triangles point out the fault occurrence, down is for the fault beginning and up is for the fault end.

As foreseen in Section 4.2, the IMF number 5 is the most informative IMF for an MDE-based diagnosis. As shown in Figure 8, the IMF 5 MDE curves present three distinct patterns in total accordance with the different experimental conditions, conversely to others’ IMFs MDE analyses. The clear sky faulty and no faulty data result in bell-shaped curves, while the faulty data generate translated curves from the clear sky data. The fault end and start curves are related to straight lines. The time segment n°4 VMD-MDE curve seems to be misplaced. Indeed, Figure 8f exhibits that time segment n°4 contains a wide variation of the output current, which explains the VMD-MDE results.

IMF number 5 is retained to process the MDE in the presented FDD method. In order to assess the VMD-MDE performance on the shading fault, two sets of data are collected and analyzed with the VMD-MDE algorithm (Figure 9 and Figure 10).

Figure 9a depicts a cloudy day with two time segments of clear sky. Figure 9b shows the MDE analysis and exhibits two different MDE curves patterns; the straight lines correspond to a cloudy sky, and the bell-shaped curves to a clear sky. One curve is misclustered, corresponding to time segment number 10. Indeed, this segment corresponds to a constant low irradiance, which induces low current production and low variations, which explain the curve trend. In further on-line application, sliding window monitoring will overcome this issue.

Figure 10 depicts a day with a clear sky, a fault occurrence and a cloudy sky after the fault. As in Figure 9b, Figure 10 reveals that clear sky curves and cloudy sky curves have different trends. In the case of a fault appearance, it can be observed that the MDE curves are translated from healthy clear sky curves as was observed in Figure 8e. However, the fault start and end curves have an equivalent trend and value in comparison to the cloudy sky curves.

In all, 36 datasets of 36 min have been collected, as shown in Table 6. These first results exhibit the links between the MDE values and the system state of health (SoH). On the basis of the previously shown curves, two scales can be retained to precisely define the PV SoH. In Figure 10, it can be observed that at scale 1, a DE value below a threshold of 0.6 depicts a cloudy sky or a fault start or end. Despite the fact that at scale 1 the DE values during a clear sky are close, whether there is faulty or no faulty data, the DE values become noticeably different at scales 4 and 5. Indeed, at scale 4, a DE value higher than 0.9 denotes a faulty state. In this work, scales 1 and 4 are chosen to determine the PV SoH with 0.6 and 0.9 as the thresholds, respectively. The overall proposed method accuracy has been evaluated at 94.29 %. In this study, the state of health is determined using a simple rule-based diagnosis, and selected scales and the associated thresholds are set from observations. In further works, more advanced classification and/or clustering methods could be investigated to enhance the presented method.

## 5. Conclusions

In this paper, an innovative online Fault Detection and Diagnosis (FDD) method has been developed. This method relies on variational mode decomposition (VMD) and multiscale dispersion entropy (MDE). The values of entropy at specific scales have been chosen as quantitative indicators to determine the state of health of the system. The VMD-MDE-based diagnosis does not require any additional equipment or sensor, can easily be implemented on-line and has a very low computation cost.

The VMD-MDE-based diagnosis capability and accuracy have been investigated on a real aging PV plant, without additional equipment, and based on the inverter output current. The first results exhibit the promising perspectives of multiscale dispersion entropy, combined with variational mode decomposition. Variational mode decomposition exposes its ability to pre-process PV data, in spite of very low frequency sampling and poor precision data. The most informative IMF has been selected and a multiscale entropy analysis has provided distinct curves trends in accordance with the experimental conditions.

The present method exclusively includes the inverter output current as the input. The information regarding the temperature, humidity, irradiance, plant structure or panel characteristics is not necessary, which ensures method robustness and scalability. The algorithms are easy to implement and require low computing capacity. Moreover, although developed under MatLab^©^, the method can be easily transposed into other programming languages, notably in Python, which make the proposed method highly adaptive.

The first promising results can be supplemented in order to produce a complete automated fault detection. Advanced classification approaches can be investigated. Moreover, in order to assess the method applicability on other faults, it will be tested on diode and hot-spot faults. The effectiveness of the method on simultaneous faults could be studied. On the assumption that the method is blind to meteorological conditions and PV plants technical characteristics, tests on different PV plant scales and technologies will be processed to confirm its robustness and scalability.

## Figures and Tables

**Figure 1 entropy-24-01311-f001:**
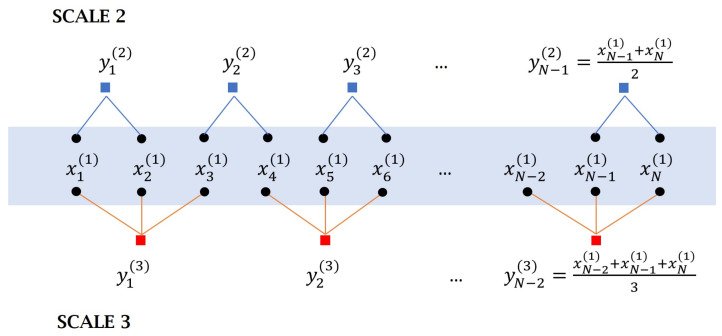
Coarse-grained process on an original time series *x* at scale factor of 2 and 3.

**Figure 2 entropy-24-01311-f002:**
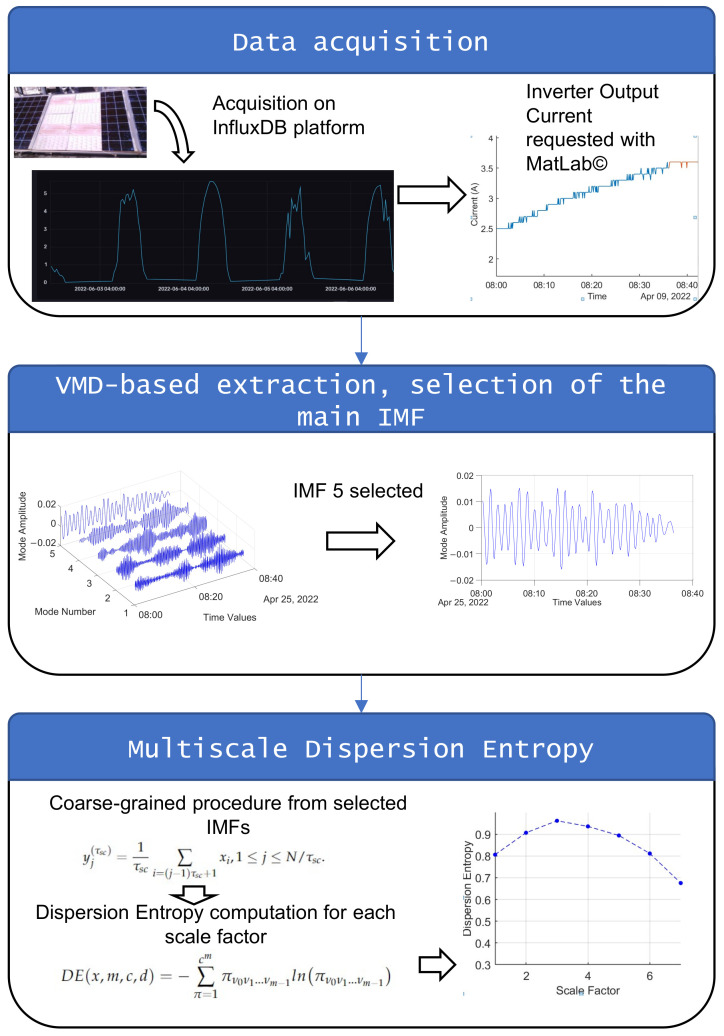
Proposed method flowchart.

**Figure 3 entropy-24-01311-f003:**
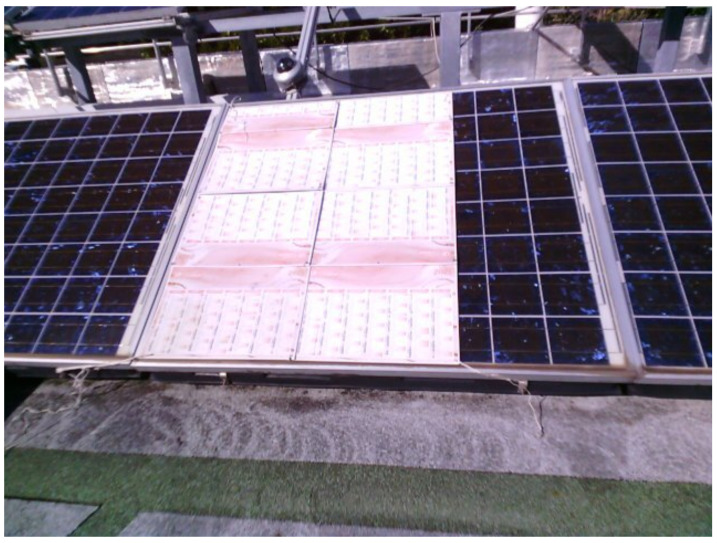
Experimental PV plant with partial shading.

**Figure 4 entropy-24-01311-f004:**
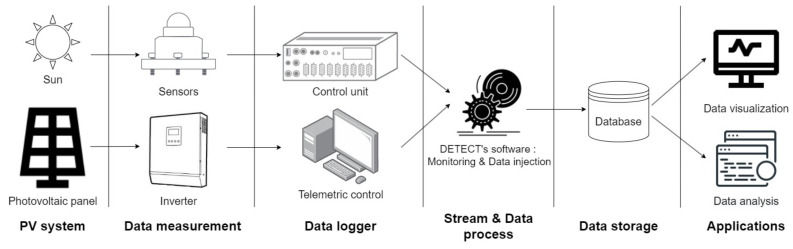
Diagram of the data pipeline.

**Figure 5 entropy-24-01311-f005:**
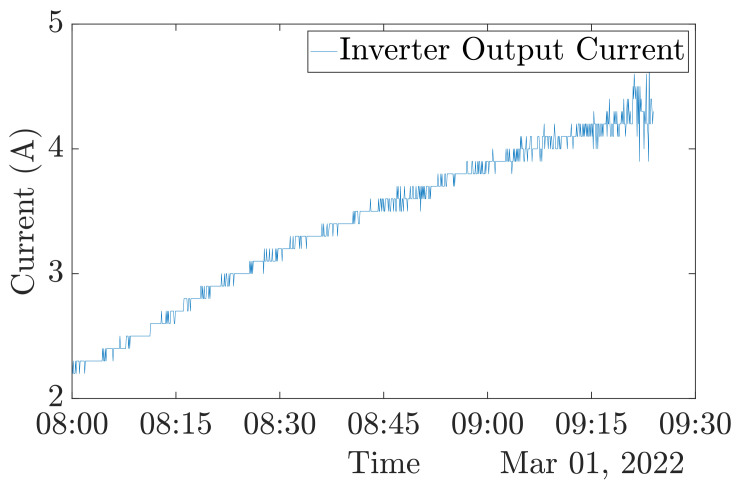
Dataset for K determination.

**Figure 6 entropy-24-01311-f006:**
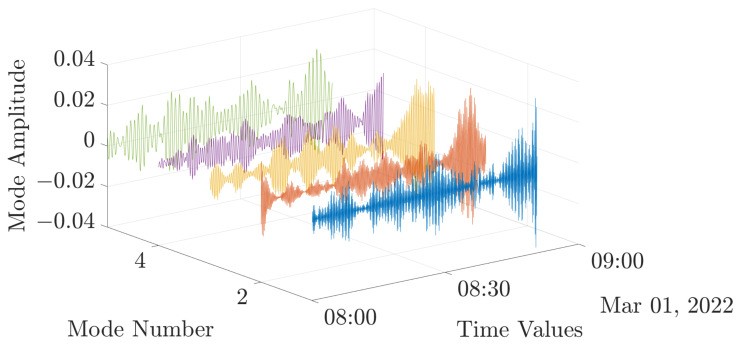
VMD of PV output current. Colors are used here in order to facilitate understanding and do not denote difference in operating conditions.

**Figure 7 entropy-24-01311-f007:**
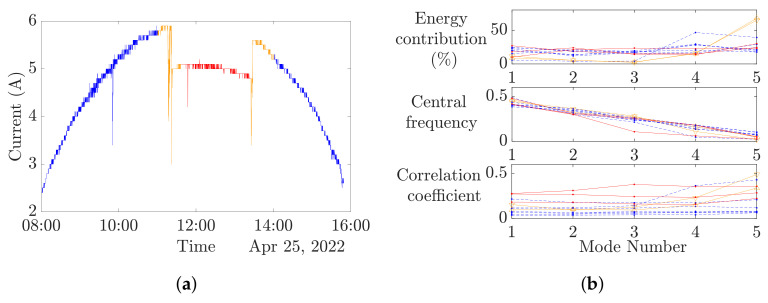
IMF Analysis. (- -) Clear sky, (-) fault, (▽) fault start, (△) fault end. (**a**) Experimental data. (**b**) IMFs analysis.

**Figure 8 entropy-24-01311-f008:**
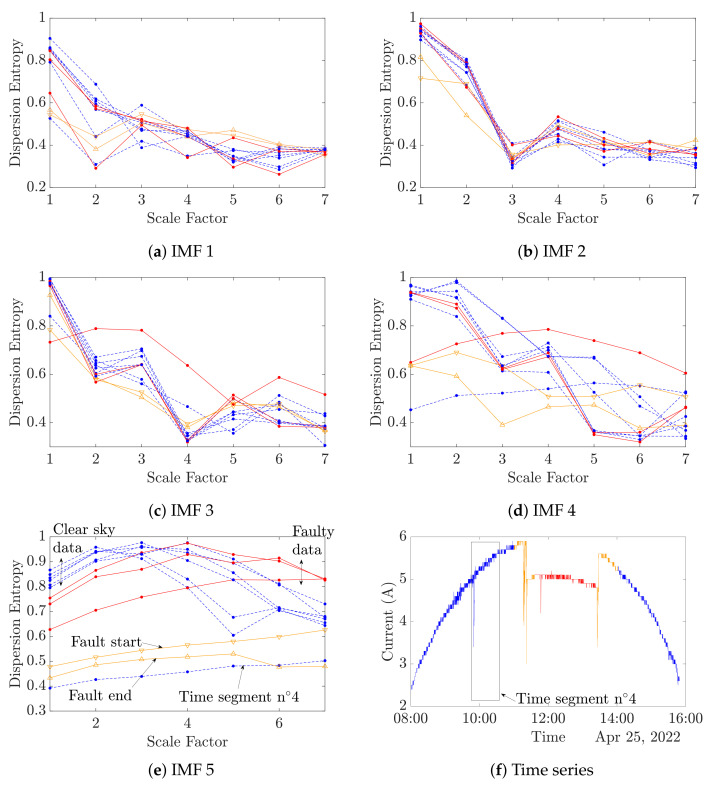
Multiscale dispersion entropy of each IMF. (- -) Clear sky, (-) fault, (▽) fault start, (△) fault end.

**Figure 9 entropy-24-01311-f009:**
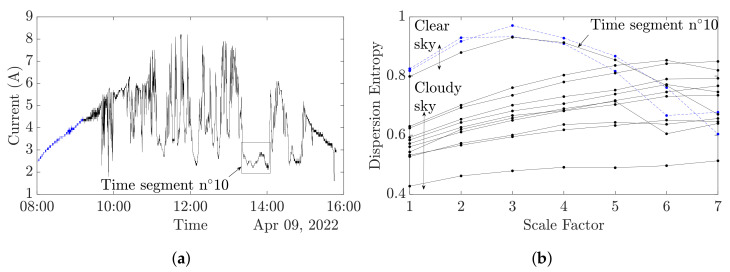
Cloudy sky conditions. (- -) Clear sky, (-) cloudy sky. (**a**) Cloudy day. (**b**) MDE analysis of cloudy day.

**Figure 10 entropy-24-01311-f010:**
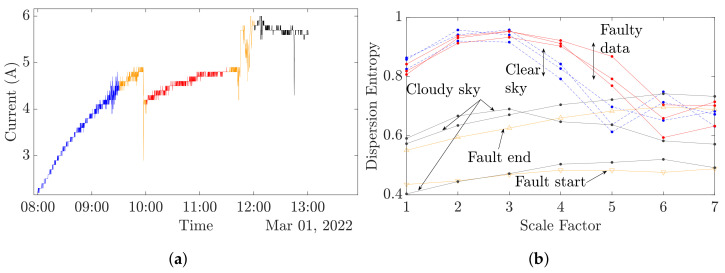
Overall conditions data. (- -) Clear sky, (-) fault, (▽) fault start, (△) fault end. (**a**) Overall conditions day. (**b**) MDE analysis of overall conditions day.

**Table 1 entropy-24-01311-t001:** TE1700/TE1500—Total Energy PV module characteristics.

Variable	Line 1 Module (TE 1500)	Line 2 Module (TE 1700)
Cell type	Poly-crystalline	
$P_{\max}$	175 Wp	170 Wp
$V_{\mathrm{oc}}$	43.8 V	43.8 V
$I_{\mathrm{sc}}$	5.3 A	5.2 A
$V_{\mathrm{mp}}$	35.7 V	35.5 V
$I_{\mathrm{mp}}$	4.9 A	4.8 A
Bypass diode number	4	

**Table 2 entropy-24-01311-t002:** Equipment installed on the PV plant.

Variable (Unit)	Sensor—Manufacturer
Incident shortwave global and diffuse irradiance (W/m^2^)	SPN1—*Delta-T Devices*
Ambient temperature (°C)	Thermocouple type K—*TC SA*
Back surface temperature of photovoltaic panel (°C)	Thermocouple type T—*TC SA*

**Table 3 entropy-24-01311-t003:** Central frequencies for different K values in healthy conditions data.

K	Frequencies							
2	0.0926	0.0211						
3	0.1036	0.0627	0.0217					
4	0.2031	0.0759	0.0427	0.0114				
5	0.2601	0.1160	0.0666	0.0432	0.0191			
6	0.3079	0.1389	0.0795	0.0458	0.0225	0.0071		
7	0.4167	0.3495	0.1266	0.0668	0.0440	0.0222	0.0070	
8	0.3759	0.2154	0.1423	0.1002	0.0653	0.0439	0.0218	0.0068

**Table 4 entropy-24-01311-t004:** Selected VMD parameters.

Parameter	Value
α	10,000
τ	0.01
*K*	5

**Table 5 entropy-24-01311-t005:** Selected MDE paramters.

Parameter	Value
*d*	1
*c*	6
*m*	2
$\tau_{sc}$	7
*N*	360

**Table 6 entropy-24-01311-t006:** Dataset description.

Experimental Condition	Number of Dataset
*All*	36
*No Fault-Clear Sky*	12
*No Fault-Cloudy Sky*	14
*Fault-Clear Sky*	10

## Data Availability

Not applicable.

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
