# Peer review of "PV System Failures Diagnosis Based on Multiscale Dispersion Entropy"

_entropy, 2022, doi:10.3390/e24091311_

Round 1
Reviewer 1 Report
The authors present a Fault Detection and Diagnosis method based on combining Variational Mode Decomposition (VMD) and Multiscale Dispersion Entropy (MDE) to detect and isolate faults occurred on a PV system. The results are somewhat interesting, but there are several deficiencies to be addressed to further proceed with this manuscript.
1. Presentation and English need to be checked carefully. For example, line 315 “it is proposed to computed…”; line 320 “is a outcome of following…”, etc.
2. The experimental conditions only contain several simple scenarios. The authors should consider more scenarios such as cloudy day without clear sky while occurring faults.
3. The authors must summarize some quantitative indicators or a clear method to detect the moments of fault start and fault end, and identify the difference between fault and shading.
4. The present work only verifies the feasibility of proposed FDD method. But its effectiveness has not been checked. Therefore, the authors should compare their method with other FDD methods to further show its promising application.
Author Response
Dear reviewer,
Thank you for agreeing to review our article. A pdf file is attached to address all your comments and suggestions.
We thank you again.
Best regards,
Fabrice KBIDI

Reviewer 2 Report
This paper proposes a fault diagnosis method for PV system using a multiscale dispersion Entropy algorithm. The article is well prepared and the descriptions are detailed. There are some points must be addressed by the authors.
1. The proposed method was applied to detect shading fault using inverter output current. However, many different faults have been described in this article from line 31 to line 45. It is more valuable if the developed method can diagnose at least 3 different faults, especially this PV has been operated for 20 years that may occur corrosion and degradation.
2. It will be more valuable (non-mandatory) if the proposed method can be applied to large PV power plants such as hundreds of kW or MW scale.
3. Please correct some typos like line 265 (2,04 kWp or 2.04 kWp?) and line 365.
Author Response

(The authors gave the same response as above.)

Reviewer 3 Report
In this paper authors have introduced an on-line diagnosis method using PV plant electrical output. I have some of the comment regarding this paper.
1. Introduction should be revised. More recent related literatures should be included.
2. Some of the experimental papers related to PV degradation should be included, i.e.,
https://www.mdpi.com/1996-1944/13/2/470,
https://www.mdpi.com/1996-1944/12/24/4047
3. The result obtained by this method should be compared with the other reported method.
4. The size of the texts of each Figure should be enlarged.
5. The “Discussion” part should be extended.
6. The ‘Conclusions’ part should be included.
7. The heading “6. Patents” should be removed.
Author Response

(The authors gave the same response as above.)

Round 2
Reviewer 1 Report
The authors have considered all the comments and the paper can be accepted in present form.
Reviewer 2 Report
The authors have responded all my questions. I have no other comments.
Reviewer 3 Report
accepted